# BernNet: Learning Arbitrary Graph Spectral Filters via Bernstein Approximation

**Mingguo He**
Renmin University of China
mingguo@ruc.edu.cn

**Zhewei Wei**[*]
Renmin University of China
zhewei@ruc.edu.cn

**Zengfeng Huang**
Fudan University
huangzf@fudan.edu.cn

**Hongteng Xu**[*]
Renmin University of China
hongtengxu@ruc.edu.cn

## Abstract

Many representative graph neural networks, *e.g.*, GPR-GNN and ChebNet, approximate graph convolutions with graph spectral filters. However, existing work either applies predefined filter weights or learns them without necessary constraints, which may lead to oversimplified or ill-posed filters. To overcome these issues, we propose *BernNet*, a novel graph neural network with theoretical support that provides a simple but effective scheme for designing and learning arbitrary graph spectral filters. In particular, for any filter over the normalized Laplacian spectrum of a graph, our BernNet estimates it by an order-$K$ Bernstein polynomial approximation and designs its spectral property by setting the coefficients of the Bernstein basis. Moreover, we can learn the coefficients (and the corresponding filter weights) based on observed graphs and their associated signals and thus achieve the BernNet specialized for the data. Our experiments demonstrate that BernNet can learn arbitrary spectral filters, including complicated band-rejection and comb filters, and it achieves superior performance in real-world graph modeling tasks. Code is available at https://github.com/ivam-he/BernNet.

## 1 Introduction

Graph neural networks (GNNs) have received extensive attention from researchers due to their excellent performance on various graph learning tasks such as social analysis [24, 17, 29], drug discovery [12, 25], traffic forecasting [18, 3, 6], recommendation system [38, 32] and computer vision [39, 4]. Recent studies suggest that many popular GNNs operate as polynomial graph spectral filters [7, 13, 5, 16, 2, 35]. Specifically, we denote an undirected graph with node set $V$ and edge set $E$ as $G = (V, E)$, whose adjacency matrix is $\mathbf{A}$. Given a signal $\mathbf{x} = [x] \in R^n$ on the graph, where $n = |V|$ is the number of nodes, we can formulate its graph spectral filtering operation as $\sum_{k=0}^{K} w_k \mathbf{L}^k \mathbf{x}$, $w_k$'s are the filter weights, $\mathbf{L} = \mathbf{I} - \mathbf{D}^{-1/2}\mathbf{A}\mathbf{D}^{-1/2}$ is the symmetric normalized Laplacian matrix of $G$, and $\mathbf{D}$ is the diagonal degree matrix of $\mathbf{A}$. Another equivalent polynomial filtering operation is $\sum_{k=0}^{K} c_k \mathbf{P}^k \mathbf{x}$, where $\mathbf{P} = \mathbf{D}^{-1/2}\mathbf{A}\mathbf{D}^{-1/2}$ is the normalized adjacency matrix and $c_k$'s are the filter weights.

---

[*]Zhewei Wei and Hongteng Xu are the corresponding authors. Work partially done at Gaoling School of Artificial Intelligence, Beijing Key Laboratory of Big Data Management and Analysis Methods, MOE Key Lab of Data Engineering and Knowledge Engineering, Renmin University of China, and Pazhou Lab, Guangzhou, 510330, China.

35th Conference on Neural Information Processing Systems (NeurIPS 2021).

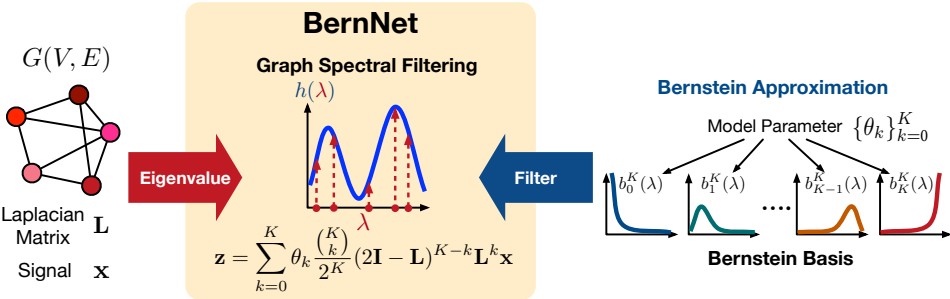
Figure 1: An illustration of the proposed BernNet.

We can broadly categorize the GNNs applying the above filtering operation into two classes, depending on whether they design the filter weights or learn them based on observed graphs. Some representative models in these two classes are shown below.

- **The GNNs driven by designing filters:** GCN [13] uses a simplified first-order Chebyshev polynomial, which is proven to be a low-pass filter [1, 31, 34, 41]. APPNP [14] utilizes Personalized PageRank (PPR) to set the filter weights and achieves a low-pass filter as well [15, 41]. GNN-LF/HF [41] designs filter weights from the perspective of graph optimization functions, which can simulate high- and low-pass filters.

- **The GNNs driven by learning filters:** ChebNet [7] approximates the filtering operation with Chebyshev polynomials, and learns a filter via trainable weights of the Chebyshev basis. GPR-GNN [5] learns a polynomial filter by directly performing gradient descent on the filter weights, which can derive high- or low-pass filters. ARMA [2] learns a rational filter via the family of Auto-Regressive Moving Average filters [21].

Although the above GNNs achieve some encouraging results in various graph modeling tasks, they still suffer from two major drawbacks. Firstly, most existing methods focus on designing or learning simple filters (*e.g.*, low- and/or high-pass filters), while real-world applications often require much more complex filters such as band-rejection and comb filters. To the best of our knowledge, none of the existing work supports designing arbitrary interpretable spectral filters. The GNNs driven by learning filters can learn arbitrary filters in theory, but they cannot intuitively show what filters they have learned. In other words, their interpretability is poor. For example, GPR-GNN [5] learns the filter weights $w_k$'s but only proves a small subset of the learnt weight sequences corresponds to low- or high-pass filters. Secondly, the GNNs often design their filters empirically or learn the filter weights without any necessary constraints. As a result, their filter weights often have poor controllability. For example, GNN-LF/HF [41] designs its filters with a complex and non-intuitive polynomial with difficult-to-tune hyperparameters. The multi-layer GCN/SGC [13, 31] leads to "ill-posed" filters (*i.e.*, those deriving negative spectral responses). Additionally, the filters learned by GPR-GNN [5] or ChebNet [7] have a chance to be ill-posed as well.

To overcome the above issues, we propose a novel graph neural network called *BernNet*, which provides an effective algorithmic framework for designing and learning arbitrary graph spectral filters. As illustrated in Figure 1, for an arbitrary spectral filter $h : [0, 2] \mapsto [0, 1]$ over the spectrum of the symmetric normalized Laplacian $\mathbf{L}$, our BernNet approximates $h$ by a $K$-order Bernstein polynomial approximation, *i.e.*, $h(\lambda) = \sum_{k=0}^{K} \theta_k b_k^K(\lambda)$. The non-negative coefficients $\{\theta_k\}_{k=0}^{K}$ of the Bernstein basis $\{b_k^K(\lambda)\}_{k=0}^{K}$ work as the model parameter, which can be interpreted as $h(2k/K)$, $k = 0, \ldots, K$ (*i.e.*, the filter values uniformly sampled from $[0, 2]$). By designing or learning the $\theta_k$'s, we can obtain various spectral filters, whose filtering operation can be formulated as $\sum_{k=0}^{K} \theta_k \frac{1}{2^K} \binom{K}{k} (2\mathbf{I} - \mathbf{L})^{K-k} \mathbf{L}^k \mathbf{x}$, where $\mathbf{x}$ is the graph signal. We further demonstrate the rationality of our BernNet from the perspective of graph optimization — any valid polynomial filers, *i.e.*, those polynomial functions mapping $[0, 2]$ to $[0, 1]$, can always be expressed by our BernNet, and accordingly, the filters learned by our BernNet are always valid. Finally, we conduct experiments to demonstrate that 1) BernNet can learn arbitrary graph spectral filters (e.g., band-rejection, comb, low-band-pass, etc.), and 2) BernNet achieves superior performance on real-world datasets.

## 2 BernNet

### 2.1 Bernstein approximation of spectral filters

Given an arbitrary filter function $h : [0, 2] \mapsto [0, 1]$, let $\mathbf{L} = \mathbf{U}\boldsymbol{\Lambda}\mathbf{U}^T$ denote the eigendecomposition of the symmetric normalized Laplacian matrix $\mathbf{L}$, where $\mathbf{U}$ is the matrix of eigenvectors and $\boldsymbol{\Lambda} = diag[\lambda_1, ..., \lambda_n]$ is the diagonal matrix of eigenvalues. We use

$$h(\mathbf{L})\mathbf{x} = \mathbf{U}h(\boldsymbol{\Lambda})\mathbf{U}^T\mathbf{x} = \mathbf{U}diag[h(\lambda_1), ..., h(\lambda_n)]\mathbf{U}^T\mathbf{x} \tag{1}$$

to denote a spectral filter on graph signal $\mathbf{x}$. The key of our work is approximate $h(\mathbf{L})$ (or, equivalently, $h(\lambda)$). For this purpose, we leverage the Bernstein basis and Bernstein polynomial approximation defined below.

**Definition 2.1** ( [10]). *(Bernstein polynomial approximation) Given an arbitrary continuous function $f(t)$ on $t \in [0, 1]$, the Bernstein polynomial approximation (of order $K$) for $f$ is defined as*

$$p_K(t) := \sum_{k=0}^{K} \theta_k \cdot b_k^K(t) = \sum_{k=0}^{K} f\left(\frac{k}{K}\right) \cdot \binom{K}{k}(1-t)^{K-k}t^k. \tag{2}$$

*Here, for $k = 0, ..., K$, $b_k^K(t) = \binom{K}{k}(1-t)^{K-k}t^k$ is the $k$-th Bernstein base, and $\theta_k = f(\frac{k}{K})$ is the function value at $k/K$, which works as the coefficient of $b_k^K(t)$.*

**Lemma 2.1** ( [10]). *Given an arbitrary continuous function $f(t)$ on $t \in [0, 1]$, let $p_K(t)$ denote the Bernstein approximation of $f(t)$ as defined in Equation (2). We have $p_K(t) \to f(t)$ as $K \to \infty$.*

For the filter function $h : [0, 2] \mapsto [0, 1]$, we let $t = \frac{\lambda}{2}$ and $f(t) = h(2t)$, so that the Bernstein polynomial approximation becomes applicable, where $\theta_k = f(k/K) = h(2k/K)$ and $b_k^K(t) = b_k^K(\frac{\lambda}{2}) = \binom{K}{k}(1 - \frac{\lambda}{2})^{K-k}(\frac{\lambda}{2})^k$ for $k = 1, ..., K$. Consequently, we can approximate $h(\lambda)$ by $p_K(\lambda/2) = \sum_{k=0}^{K} \theta_k \binom{K}{k}(1 - \frac{\lambda}{2})^{K-k}\left(\frac{\lambda}{2}\right)^k = \sum_{k=0}^{K} \theta_k \frac{1}{2^K}\binom{K}{k}(2 - \lambda)^{K-k}\lambda^k$, and Lemma 2.1 ensures that $p_K(\lambda/2) \to h(\lambda)$ as $K \to \infty$.

Replacing $\{h(\lambda_i)\}_{i=1}^n$ with $\{p_K(\lambda_i/2)\}_{i=1}^n$, we approximate the spectral filter $h(\mathbf{L})$ in Equation (1) as $\mathbf{U}diag[p_K(\lambda_1/2), ..., p_K(\lambda_n/2)]\mathbf{U}^T$ and derive the proposed BernNet. In particular, given a graph signal $\mathbf{x}$, the convolutional operator of our BernNet is defined as follows:

$$\mathbf{z} = \underbrace{\mathbf{U}diag[p_K(\lambda_1/2), ..., p_K(\lambda_n/2)]\mathbf{U}^T}_{\text{BernNet}}\mathbf{x} = \sum_{k=0}^{K} \theta_k \frac{1}{2^K}\binom{K}{k}(2\mathbf{I} - \mathbf{L})^{K-k}\mathbf{L}^k\mathbf{x} \tag{3}$$

where each coefficient $\theta_k$ can be either set to $h(2k/K)$ to approximate a predetermined filter $h$, or learnt from the graph structure and signal in an end-to-end fashion. As a natural extension of Lemma 2.1, our BernNet owns the following proposition.

**Proposition 2.1.** *For an arbitrary continuous filter function $h : [0, 2] \to [0, 1]$, by setting $\theta_k = h(2k/K), k = 0, \ldots, K$, the $\mathbf{z}$ in Equation (3) satisfies $\mathbf{z} \to h(\mathbf{L})\mathbf{x}$ as $K \to \infty$.*

*Proof.* According to the above derivation, we have $p_K(\lambda/2) = \sum_{k=0}^{K} \theta_k \binom{K}{k}(1 - \frac{\lambda}{2})^{K-k}\left(\frac{\lambda}{2}\right)^k = \sum_{k=0}^{K} \theta_k \frac{1}{2^K}\binom{K}{k}(2 - \lambda)^{K-k}\lambda^k$, and Lemma 2.1 ensures that $p_K(\lambda/2) \to h(\lambda)$ as $\theta_k = h(2k/K)$ and $K \to \infty$.

Consequently, we have

$$\mathbf{z} = \mathbf{U}diag[p_K(\lambda_1/2), ..., p_K(\lambda_n/2)]\mathbf{U}^T\mathbf{x} \to \mathbf{U}diag[h(\lambda_1), ..., h(\lambda_n)]\mathbf{U}^T\mathbf{x} = h(\mathbf{L})$$

as $\theta_k = h(2k/K)$ and $K \to \infty$.

$\square$

### 2.2 Realizing existing filters with BernNet.

As shown in Proposition 2.1, our BernNet can approximate arbitrary continuous spectral filters with sufficient precision. Below we give some representative examples of how our BernNet exactly realizes existing filters that are commonly used in GNNs.

Table 1: Realizing commonly used filters with BernNet.

| Filter types | Filter $h(\lambda)$ | $\theta_k$ for $k = 0, \ldots, K$ | Bernstein approximation $p_K(\frac{\lambda}{2})$ | BernNet |
|---|---|---|---|---|
| All-pass | 1 | $\theta_k = 1$ | 1 | $\mathbf{I}$ |
| Linear low-pass | $1 - \lambda/2$ | $\theta_k = 1 - k/K$ | $1 - \lambda/2$ | $\mathbf{I} - \frac{1}{2}\mathbf{L}$ |
| Linear high-pass | $\lambda/2$ | $\theta_k = k/K$ | $\lambda/2$ | $\frac{1}{2}\mathbf{L}$ |
| Impulse low-pass | $\delta_0(\lambda)$ | $\theta_0 = 1$ and other $\theta_k = 0$ | $(1 - \lambda/2)^K$ | $\frac{1}{2^K}(2\mathbf{I} - \mathbf{L})^K$ |
| Impulse high-pass | $\delta_2(\lambda)$ | $\theta_K = 1$ and other $\theta_k = 0$ | $(\lambda/2)^K$ | $\frac{1}{2^K}\mathbf{L}^K$ |
| Impulse band-pass | $\delta_1(\lambda)$ | $\theta_{K/2} = 1$ and other $\theta_k = 0$ | $\binom{K}{K/2}(1 - \lambda/2)^{K/2}(\lambda/2)^{K/2}$ | $\frac{1}{2^K}\binom{K}{K/2}(2\mathbf{I} - \mathbf{L})^{K/2}\mathbf{L}^{K/2}$ |

- **All-pass filter** $h(\lambda) = 1$. We set $\theta_k = 1$ for $k = 0, \ldots, K$, and the approximation $p_K(\frac{\lambda}{2}) = 1$ is exactly the same with $h(\lambda)$. Accordingly, our BernNet becomes an identity matrix, which realizes the all-pass filter perfectly.

- **Linear low-pass filter** $h(\lambda) = 1 - \lambda/2$. We set $\theta_k = 1 - k/K$ for $k = 0, \ldots, K$ and obtain $p_K(\frac{\lambda}{2}) = 1 - \lambda/2$. The BernNet becomes $\sum_{k=0}^{K} \frac{(K-k)}{K}\frac{1}{2^K}\binom{K}{k}(2\mathbf{I} - \mathbf{L})^{K-k}\mathbf{L}^k = \mathbf{I} - \frac{1}{2}\mathbf{L}$, which achieves the linear low-pass filter exactly. Note that $\mathbf{I} - \frac{1}{2}\mathbf{L} = \frac{1}{2}(\mathbf{I} + \mathbf{P})$ is also the same as the graph convolutional network (GCN) before renormalization [13].

- **Linear high-pass filter** $h(\lambda) = \lambda/2$. Similarly, we can set $\theta_k = k/K$ for $k = 0, \ldots, K$ to get a perfect approximation $p_K(\frac{\lambda}{2}) = \frac{\lambda}{2}$, and the BernNet becomes $\frac{1}{2}\mathbf{L}$.

Note that even for those non-continuous spectral filters, $e.g.$, the impulse low/high/band-pass filters, our BernNet can also provide good approximations (with sufficient large $K$).

- **Impulse low-pass filter** $h(\lambda) = \delta_0(\lambda)$.[†] We set $\theta_0 = 1$ and $\theta_k = 0$ for $k \neq 0$, and $p_K(\frac{\lambda}{2}) = (1 - \frac{\lambda}{2})^K$. Accordingly, the BernNet becomes $\frac{1}{2^K}(2\mathbf{I} - \mathbf{L})^K$, deriving an $K$-layer linear low-pass filter.

- **Impulse high-pass filter** $h(\lambda) = \delta_2(\lambda)$. We set $\theta_K = 1$ and $\theta_k = 0$ for $k \neq K$, and $p_K(\frac{\lambda}{2}) = (\frac{\lambda}{2})^K$. The BernNet becomes $\frac{1}{2^K}\mathbf{L}^K$, $i.e.$, an $K$-layer linear high-pass filter.

- **Impulse band-pass filter** $h(\lambda) = \delta_1(\lambda)$. Similarly, we set $\theta_{K/2} = 1$ and $\theta_k = 0$ for $k \neq K/2$, and $p_K(\frac{\lambda}{2}) = \binom{K}{K/2}(1 - \lambda/2)^{K/2}(\lambda/2)^{K/2}$. The BernNet becomes $\frac{1}{2^K}\binom{K}{K/2}(2\mathbf{I} - \mathbf{L})^{K/2}\mathbf{L}^{K/2}$, which can be explained as stacking a $K/2$-layer linear low-pass filter and a $K/2$-layer linear high-pass filter. Obviously, $K$ should be an even number in this case.

Table 1 summarizes the design of the BernNet for the filters above. We can find that an appealing advantage of our BernNet is that its coefficients are highly correlated with the spectral property of the target filter. In particular, we can determine to pass or reject the spectral signal with $\lambda \approx \frac{2k}{K}$ by using a large or small $\theta_k$ because each Bernstein base $b_k^K(\lambda)$ corresponds to a "bump" located at $\frac{2k}{K}$. This property provides useful guidance when designing filters, which enhances the interpretability of our BernNet.

## 2.3 Learning complex filters with BernNet

Besides designing the above typical filters, our BernNet can express more complex filters, such as band-pass, band-rejection, comb, low-band-pass filters, $etc$. Moreover, given the graph signals before and after applying such filters ($i.e.$, the $\mathbf{x}$'s and the corresponding $\mathbf{z}$'s), our BernNet can learn their approximations in an end-to-end manner. Specifically, given the pairs $\{\mathbf{x}, \mathbf{z}\}$, we learn the coefficients $\{\theta_k\}_{k=0}^K$ of the BernNet by gradient descent. More implementation details can be found at the experimental section below. Figure 2 illustrates the four complex filters and the approximations we learned (The low-band pass filter is $h(\lambda) = I_{[0,0.5]}(\lambda) + \exp\left(-100(\lambda - 0.5)^2\right)I_{(0.5,1)}(\lambda) + \exp\left(-50(\lambda - 1.5)^2\right)I_{[1,2]}(\lambda)$, where $I_\Omega(\lambda) = 1$ when $\lambda \in \Omega$, otherwise $I_\Omega(\lambda) = 0$). In general, our BernNet can learn a smoothed approximation of these complex filters, and the approximation precision improves with the increase of the order $K$. Note that although the BernNet cannot pinpoint the exact peaks of the comb filter or drop to 0 for the valleys of comb or low-band-pass filters due to the limitation of $K$, it still significantly outperforms other GNNs for learning such complex filters.

---

[†]The impulse function $\delta_x(\lambda) = 1$ if $\lambda = x$, otherwise $\delta_x(\lambda) = 0$

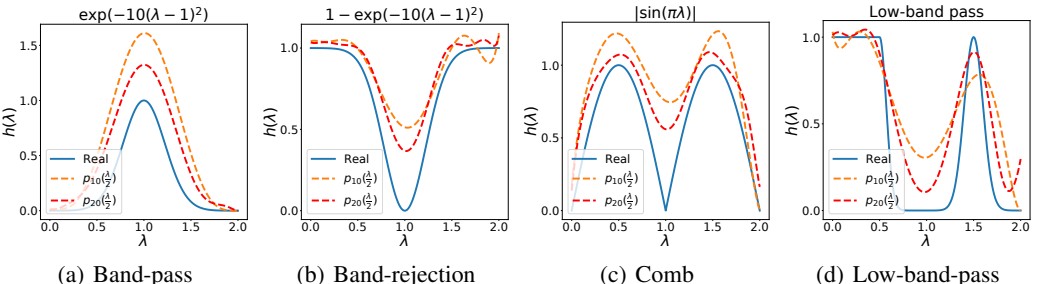

Figure 2: Illustrations of four complex filters and their approximations learnt by BernNet.

# 3 BernNet in the Lens of Graph Optimization

In this section, we motivate BernNet from the perspective of graph optimization. In particular, we show that any polynomial filter that attempts to approximate a valid filter has to take the form of BernNet.

## 3.1 A generalized graph optimization problem

Given a $n$-dimensional graph signal $\mathbf{x}$, we consider a generalized graph optimization problem

$$\min_{\mathbf{z}} f(\mathbf{z}) = (1-\alpha)\mathbf{z}^T \gamma(\mathbf{L})\mathbf{z} + \alpha\|\mathbf{z} - \mathbf{x}\|_2^2 \tag{4}$$

where $\alpha \in [0,1)$ is a trade-off parameter, $\mathbf{z} \in R^n$ denotes the propagated representation of the input graph signal $\mathbf{x}$, and $\gamma(\mathbf{L})$ denotes an energy function of $\mathbf{L}$, determining the rate of propagation [28]. Generally, $\gamma(\cdot)$ operates on the spectral of $\mathbf{L}$, and we have $\gamma(\mathbf{L}) = \mathbf{U}diag[\gamma(\lambda_1), ..., \gamma(\lambda_n)]\mathbf{U}^T$.

We can model the polynomial filtering operation of existing GNNs with the optimal solution of Equation (4). For example, if we set $\gamma(\mathbf{L}) = \mathbf{L}$, then the optimization function (4) becomes $f(\mathbf{z}) = (1-\alpha)\mathbf{z}^T\mathbf{L}\mathbf{z} + \alpha\|\mathbf{z}-\mathbf{x}\|_2^2$, a well-known convex graph optimization function proposed by Zhou et al. [40]. $f(\mathbf{z})$ takes the minimum when the derivative $\frac{\partial f(\mathbf{z})}{\partial \mathbf{z}} = 2(1-\alpha)\mathbf{L}\mathbf{z} + 2\alpha\,(\mathbf{z}-\mathbf{x}) = \mathbf{0}$, which solves to

$$\mathbf{z}^* = \alpha\left(\mathbf{I} - (1-\alpha)(\mathbf{I}-\mathbf{L})\right)^{-1}\mathbf{x} = \sum_{k=0}^{\infty} \alpha(1-\alpha)^k\left(\mathbf{I}-\mathbf{L}\right)^k\mathbf{x} = \sum_{k=0}^{\infty} \alpha(1-\alpha)^k\mathbf{P}^k\mathbf{x}.$$

By taking a suffix sum $\sum_{k=0}^{K} \alpha(1-\alpha)^k\mathbf{P}^k\mathbf{x}$, we obtain the polynomial filtering operation for APPNP [14]. Zhu et al. [41] further show that GCN [13], DAGNN [19], and JKNet [36] can be interpreted by the optimization function (4) with $\gamma(\mathbf{L}) = \mathbf{L}$.

The generalized form of Equation (4) allows us to simulate more complex polynomial filtering operation. For example, let $\alpha = 0.5$ and $\gamma(\mathbf{L}) = e^{t\mathbf{L}} - \mathbf{I}$, a heat kernel with $t$ as the temperature parameter. Then $f(\mathbf{z})$ takes the minimum when the derivative $\frac{\partial f(\mathbf{z})}{\partial \mathbf{z}} = \left(e^{t\mathbf{L}} - \mathbf{I}\right)\mathbf{z} + \mathbf{z} - \mathbf{x} = \mathbf{0}$, which solves to

$$\mathbf{z}^* = e^{-t\mathbf{L}}\mathbf{x} = e^{-t(\mathbf{I}-\mathbf{P})}\mathbf{x} = \sum_{k=0}^{\infty} e^{-t}\frac{t^k}{k!}\mathbf{P}^k\mathbf{x}.$$

By taking a suffix sum $\sum_{k=0}^{K} e^{-t}\frac{t^k}{k!}\mathbf{P}^k\mathbf{x}$, we obtain the polynomial filtering operation for the heat kernal based GNN such as GDC [15] and GraphHeat [34].

## 3.2 Non-negative constraint on polynomial filters

A natural question is that, does an arbitrary energy function $\gamma(\mathbf{L})$ correspond to a valid or ill-posed spectral filter? Conversely, does any polynomial filtering operation $\sum_{k=0}^{K} w_k \mathbf{L}^k \mathbf{x}$ correspond to the optimal solution of the optimization function (4) for some energy function $\gamma(\mathbf{L})$?

As it turns out, there is a "minimum requirement" for the energy function $\gamma(\mathbf{L})$; $\gamma(\mathbf{L})$ has to be **positive semidefinite**. In particular, if $\gamma(\mathbf{L})$ is not positive semidefinite, then the optimization

function $f(\mathbf{z})$ is not convex, and the solution to $\frac{\partial f(\mathbf{z})}{\partial \mathbf{z}} = 0$ may corresponds to a saddle point. Furthermore, without the positive semidefinite constraint on $\gamma(\mathbf{L})$, $f(\mathbf{z})$ may goes to $-\infty$ as we set $\mathbf{z}$ to be a multiple of the eigenvector corresponding to the negative eigenvalue.

**Non-negative polynomial filters.** Given a positive semidefinite energy function $\gamma(\mathbf{L})$, we now consider how the corresponding polynomial filtering operation $\sum_{k=0}^{K} w_k \mathbf{L}^k \mathbf{x}$ should look like. Recall that we assume $\gamma(\mathbf{L}) = \mathbf{U} diag[\gamma(\lambda_1), ..., \gamma(\lambda_n)]\mathbf{U}^T$. By the positive semidefinite constraint, we have $\gamma(\lambda) \geq 0$ for $\lambda \in [0, 2]$. Since the objective function $f(\mathbf{z})$ is convex, it takes the minimum when $\frac{\partial f(\mathbf{z})}{\partial \mathbf{z}} = 2(1 - \alpha)\gamma(\mathbf{L})\mathbf{z} + 2\alpha (\mathbf{z} - \mathbf{x}) = \mathbf{0}$. Accordingly, the optimum $\mathbf{z}^*$ can be derived as

$$\alpha \left(\alpha\mathbf{I} + (1 - \alpha)\gamma(\mathbf{L})\right)^{-1} \mathbf{x} = \mathbf{U} diag \left[\frac{\alpha}{\alpha + (1 - \alpha)\gamma(\lambda_1)}, ..., \frac{\alpha}{\alpha + (1 - \alpha)\gamma(\lambda_n)}\right] \mathbf{U}^T\mathbf{x}. \quad (5)$$

Let $h(\lambda) = \frac{\alpha}{\alpha + (1-\alpha)\gamma(\lambda)}$ denote the exact spectral filter, and $g(\lambda) = \sum_{k=0}^{K} w_k \lambda^k$ denote a polynomial approximation of $h(\lambda)$ (e.g. the suffix sum of $h(\lambda)$'s taylor expansion). Since $\gamma(\lambda) \geq 0$ when $\lambda \in [0, 2]$, we have $0 \leq h(\lambda) \leq \frac{\alpha}{\alpha + (1-\alpha)\cdot 0} = 1$ for $\lambda \in [0, 2]$. Consequently, it is natural to assume the polynomial filter $g(\lambda) = \sum_{k=0}^{K} w_k \lambda^k$ also satisfies $0 \leq g(\lambda) \leq 1$.

**Constraint 3.1.** *Assuming the energy function $\gamma(\mathbf{L})$ is positive semidefinite, a polynomial filter $g(\lambda) = \sum_{k=0}^{K} w_k \lambda^k$ approximating the optimal solution to Equation* (4) *has to satisfy*

$$0 \leq g(\lambda) = \sum_{k=0}^{K} w_k \lambda^k \leq 1, \ \forall \ \lambda \in [0, 2]. \quad (6)$$

While Constraint 3.1 seems to be simple and intuitive, some of the existing GNN may not satisfies this constraint. For example, GCN uses $\mathbf{z} = \mathbf{P}\mathbf{x} = (\mathbf{I} - \mathbf{L})\mathbf{x}$, which corresponds to a polynomial filter $g(\lambda) = 1 - \lambda$ that takes negative value when $\lambda > 1$, violating Constraint 3.1. As shown in [31], the renormalization trick $\tilde{\mathbf{P}} = (\mathbf{I} + \mathbf{D})^{-1/2} (\mathbf{I} + \mathbf{A}) (\mathbf{I} + \mathbf{D})^{-1/2}$ shrinks the spectral and thus reliefs the problem. However, $g(\lambda)$ may still take negative value as the maximum eigenvalue of $\tilde{\mathbf{L}} = \mathbf{I} - \tilde{\mathbf{P}}$ is still larger than 1.

## 3.3 Non-negative polynomials and Bernstein basis

Constraint 3.1 motivates us to design polynomial filters $g(\lambda) = \sum_{k=0}^{K} w_k \lambda^k$ such that $0 \leq g(\lambda) \leq 1$ when $\lambda \in [0, 2]$. The $g(\lambda) \leq 1$ part is trivial, as we can always rescale each $w_k$ by a factor of $\sum_{k=0}^{K} |w_k| 2^k$. The $g(\lambda) \geq 0$ part, however, requires more elaboration. Note that we can not simply set $w_k \geq 0$ for each $k = 0 \dots, K$, since it is shown in [5] that such polynomials only correspond to low-pass filters.

As it turns out, the Bernstein basis has the following nice property: a polynomial that is non-negative on a certain interval can always be expressed as a non-negative linear combination of Bernstein basis. Specifically, we have the following lemma.

**Lemma 3.1** ([23]). *Assume a polynomial $p(x) = \sum_{k=0}^{K} \theta_k x^k$ satisfies $p(x) \geq 0$ for $x \in [0, 1]$. Then there exists a sequence of non-negative coefficients $\theta_k$, $k = 0, \dots, K$, such that*

$$p(x) = \sum_{k=0}^{K} \theta_k b_k^K(x) = \sum_{k=0}^{K} \theta_k \binom{K}{k} (1 - x)^{K-k} x^k$$

Lemma 3.1 suggests that to approximate a valid filter, the polynomial filter $g(\lambda)$ has to be a non-negative linear combination of Bernstein basis. Specifically, by setting $x = \lambda/2$, the filter $g(\lambda)$ that satisfies $g(\lambda) \geq 0$ for $\lambda \in [0, 2]$ can be expressed as

$$g(\lambda) := p\left(\frac{\lambda}{2}\right) = \sum_{k=0}^{K} \theta_k \frac{1}{2^K} \binom{K}{k} (2 - \lambda)^{K-k} \lambda^k.$$

Consequently, any valid polynomial filter that approximate the optimal solution of (4) with positive semidefinite energy function $\gamma(\mathbf{L})$ has to take the following form: $\mathbf{z} = \sum_{k=0}^{K} \theta_k \frac{1}{2^K} \binom{K}{k} (2\mathbf{I} - \mathbf{L})^{K-k} \mathbf{L}^k \mathbf{x}$. This observation motivates our BernNet from the perspective of graph optimization — any valid polynomial filers, $i.e.$, the $g : [0, 2] \mapsto [0, 1]$, can always be expressed by BernNet, and accordingly, the filters learned by our BernNet are always valid.

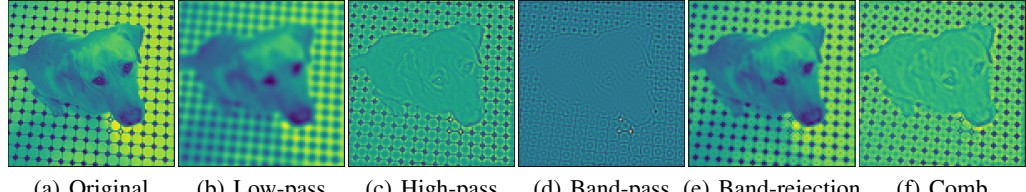

| (a) Original | (b) Low-pass | (c) High-pass | (d) Band-pass | (e) Band-rejection | (f) Comb |

Figure 3: A input image and the filtering results.

Table 2: Average sum of squared error and $R^2$ score in parentheses.

|  | Low-pass | High-pass | Band-pass | Band-rejection | Comb |
|---|---|---|---|---|---|
|  | $\exp(-10\lambda^2)$ | $1 - \exp(-10\lambda^2)$ | $\exp(-10(\lambda-1)^2)$ | $1 - \exp(-10(\lambda-1)^2)$ | $|\sin(\pi\lambda)|$ |
| GCN | 3.4799(.9872) | 67.6635(.2364) | 25.8755(.1148) | 21.0747(.9438) | 50.5120(.2977) |
| GAT | 2.3574(.9905) | 21.9618(.7529) | 14.4326(.4823) | 12.6384(.9652) | 23.1813(.6957) |
| GPR-GNN | 0.4169(.9984) | 0.0943(.9986) | 3.5121(.8551) | 3.7917(.9905) | 4.6549(.9311) |
| ARMA | 1.8478(.9932) | 1.8632(.9793) | 7.6922(.7098) | 8.2732(.9782) | 15.1214(.7975) |
| ChebNet | 0.8220(.9973) | 0.7867(.9903) | 2.2722(.9104) | 2.5296(.9934) | 4.0735(.9447) |
| BernNet | **0.0314(.9999)** | **0.0113(.9999)** | **0.0411(.9984)** | **0.9313(.9973)** | **0.9982(.9868)** |

## 4  Related Work

Graph neural networks (GNNs) can be broadly divided into spectral-based GNNs and spatial-based GNNs [33].

Spectral-based GNNs design spectral graph filters in the spectral domain. ChebNet [7] uses Chebyshev polynomial to approximate a filter. GCN [13] simplifies the Chebyshev filter with the first-order approximation. GraphHeat [34] uses heat kernel to design a graph filter. APPNP [14] utilizes Personalized PageRank (PPR) to set the filter weights. GPR-GNN [5] learns the polynomial filters via gradient descent on the polynomial coefficients. ARMA [2] learns a rational filter via the family of Auto-Regressive Moving Average filters [21]. AdaGNN [9] learns simple filters across multiple layers with a single parameter for each feature channel at each layer. As aforementioned, these methods mainly focus on designing low- or high-pass filters or learning filters without any constraints, which may lead to misspecified even ill-posed filters.

On the other hand, spatial-based GNNs directly propagate and aggregate graph information in the spatial domain. From this perspective, GCN [13] can be explained as the aggregation of the one-hop neighbor information on the graph. GAT [30] uses the attention mechanism to learn aggregation weights. Recently, Balcilar et al. [1] bridge the gap between spectral-based and spatial-based GNNs and unify GNNs in the same framework. Their work shows that the GNNs can be interpreted as sophisticated data-driven filters. This motivates the design of the proposed BernNet, which can learn arbitrary non-negative spectral filters from real-world graph signals.

## 5  Experiments

In this section, we conduct experiments to evaluate BernNet's capability to learn arbitrary filters as well as the performance of BernNet on real datasets. All the experiments are conducted on a machine with an NVIDIA TITAN V GPU (12GB memory), Intel Xeon CPU (2.20 GHz), and 512GB of RAM.

### 5.1  Learning filters from the signal

We conduct an empirical analysis on 50 real images with the resolution of 100×100 from the Image Processing Toolbox in Matlab. We conduct independent experiments on these 50 images and report the average of the evaluation index. Following the experimental setting in [1], we regard each image as a 2D regular 4-neighborhood grid graph. The graph structure translates to an $10,000 \times 10,000$ adjacency matrix while the pixel intensity translates to a $10,000$-dimensional signal vector.

Table 3: Dataset statistics.

|          | Cora | CiteSeer | PubMed | Computers | Photo  | Chameleon | Squirrel | Actor | Texas | Cornell |
|----------|------|----------|--------|-----------|--------|-----------|----------|-------|-------|---------|
| Nodes    | 2708 | 3327     | 19717  | 13752     | 7650   | 2277      | 5201     | 7600  | 183   | 183     |
| Edges    | 5278 | 4552     | 44324  | 245861    | 119081 | 31371     | 198353   | 26659 | 279   | 277     |
| Features | 1433 | 3703     | 500    | 767       | 745    | 2325      | 2089     | 932   | 1703  | 1703    |
| Classes  | 7    | 6        | 5      | 10        | 8      | 5         | 5        | 5     | 5     | 5       |

For each of the 50 images, we apply 5 different filters (low-pass, high-pass, band-pass, band-rejection and comb) to the spectral domain of its signal. The formula of each filter is shown in Table 2. Recall that applying a low-pass filter $\exp(-10\lambda^2)$ to the spectral domain $\mathbf{L} = \mathbf{U}diag\left[\lambda_1, \ldots, \lambda_n\right]\mathbf{U}^\top$ means applying $\mathbf{U}diag\left[\exp(-10\lambda_1^2), \ldots, \exp(-10\lambda_n^2)\right]\mathbf{U}^\top$ to the graph signal. Figure 3 shows the one of the input image and the corresponding filtering results.

In this task, we use the original graph signal as the input and the filtering signal to supervise the training process. The goal is to minimize the square error between output and the filtering signal by learning the correct filter. We evaluate BernNet against five popular GNN models: GCN [13], GAT [30], GPR-GNN [5], ARMA [2] and ChebNet [7]. To ensure fairness, we use two convolutional units and a linear output layer for all models. We train all models with approximately 2k trainable parameters and tune the hidden units to ensure they have similar parameters. Following [1], we discard any regularization or dropout and simply force the GNN to learn the input-output relation. For all models, we set the maximum number of epochs to 2000 and stop the training if the loss does not drop for 100 consecutive times and use Adam optimization with a 0.01 learning rate without decay. Models do not use the position information of the picture pixels. We use a mask to cover the edge nodes of the picture, so the problem can be regarded as a simple regression problem. For BernNet, we use a two-layer model, with each layer sharing the same set of $\theta_k$ for $k = 0, \ldots, K$ and set $K = 10$. For GPR-GNN, we use the officially released code (see the supplementary materials for URL and commit numbers) and set the order of polynomial filter $K = 10$. Other baseline models are based on Pytorch Geometric implementation [11]. The more detailed experiments setting can be found in the Appendix.

Table 2 shows the average of the sum of squared error (lower the better) and the $R^2$ scores (higher the better). We first observe that GCN and GAT can only handle low-pass filters, which concurs with the theoretical analysis in [1]. GPR-GNN, ARMA and ChebNet can learn different filters from the signals. However, BernNet consistently outperformed these models by a large margin on all tasks in terms of both metrics. We attribute this quality to BernNet's ability to tune the coefficients $\theta_k$'s, which directly correspond to the uniformly sampled filter values.

## 5.2 Node classification on real-world datasets

We now evaluate the performance of BernNet against the competitors on real-world datasets. Following [5], we include three citation graph Cora, CiteSeer and PubMed [27, 37], and the Amazon co-purchase graph Computers and Photo [20]. As shown in [5] these 5 datasets are homophilic graphs on which the connected nodes tend to share the same label. We also include the Wikipedia graph Chameleon and Squirrel [26], the Actor co-occurrence graph, and webpage graphs Texas and Cornell from WebKB[‡] [22]. These 5 datasets are heterophilic datasets on which connected nodes tend to have different labels. We summarize the statistics of these datasets in Table 3.

Following [5], we perform full-supervised node classification task with each model, where we randomly split the node set into train/validation/test set with ratio 60%/20%/20%. For fairness, we generate 10 random splits by random seeds and evaluate all models on the same splits, and report the average metric for each model.

We compare BernNet with 6 baseline models: MLP, GCN [13], GAT [30], APPNP [14], ChebNet [7], and GPR-GNN [5]. For GPR-GNN, we use the officially released code (see the supplementary materials for URL and commit numbers) and set the order of polynomial filter $K = 10$. For other models, we use the corresponding Pytorch Geometric library implementations [11]. For BernNet, we

---

[‡] http://www.cs.cmu.edu/afs/cs.cmu.edu/project/theo-11/www/wwkb/

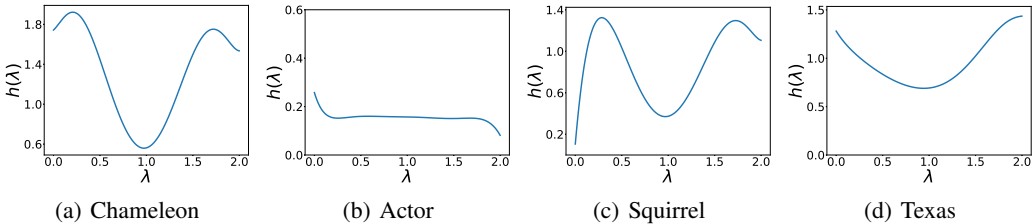

|  | (a) Chameleon | (b) Actor | (c) Squirrel | (d) Texas |

Figure 4: Filters learnt from real-world datasets by BernNet.

Table 4: Results on real world benchmark datasets: Mean accuracy (%) ± 95% confidence interval.

|  | GCN | GAT | APPNP | MLP | ChebNet | GPR-GNN | BernNet |
|---|---|---|---|---|---|---|---|
| Cora | $87.14_{\pm1.01}$ | $88.03_{\pm0.79}$ | $88.14_{\pm0.73}$ | $76.96_{\pm0.95}$ | $86.67_{\pm0.82}$ | $\mathbf{88.57_{\pm0.69}}$ | $88.52_{\pm0.95}$ |
| CiteSeer | $79.86_{\pm0.67}$ | $\mathbf{80.52_{\pm0.71}}$ | $80.47_{\pm0.74}$ | $76.58_{\pm0.88}$ | $79.11_{\pm0.75}$ | $80.12_{\pm0.83}$ | $80.09_{\pm0.79}$ |
| PubMed | $86.74_{\pm0.27}$ | $87.04_{\pm0.24}$ | $88.12_{\pm0.31}$ | $85.94_{\pm0.22}$ | $87.95_{\pm0.28}$ | $88.46_{\pm0.33}$ | $\mathbf{88.48_{\pm0.41}}$ |
| Computers | $83.32_{\pm0.33}$ | $83.32_{\pm0.39}$ | $85.32_{\pm0.37}$ | $82.85_{\pm0.38}$ | $87.54_{\pm0.43}$ | $86.85_{\pm0.25}$ | $\mathbf{87.64_{\pm0.44}}$ |
| Photo | $88.26_{\pm0.73}$ | $90.94_{\pm0.68}$ | $88.51_{\pm0.31}$ | $84.72_{\pm0.34}$ | $93.77_{\pm0.32}$ | $\mathbf{93.85_{\pm0.28}}$ | $93.63_{\pm0.35}$ |
| Chameleon | $59.61_{\pm2.21}$ | $63.13_{\pm1.93}$ | $51.84_{\pm1.82}$ | $46.85_{\pm1.51}$ | $59.28_{\pm1.25}$ | $67.28_{\pm1.09}$ | $\mathbf{68.29_{\pm1.58}}$ |
| Actor | $33.23_{\pm1.16}$ | $33.93_{\pm2.47}$ | $39.66_{\pm0.55}$ | $40.19_{\pm0.56}$ | $37.61_{\pm0.89}$ | $39.92_{\pm0.67}$ | $\mathbf{41.79_{\pm1.01}}$ |
| Squirrel | $46.78_{\pm0.87}$ | $44.49_{\pm0.88}$ | $34.71_{\pm0.57}$ | $31.03_{\pm1.18}$ | $40.55_{\pm0.42}$ | $50.15_{\pm1.92}$ | $\mathbf{51.35_{\pm0.73}}$ |
| Texas | $77.38_{\pm3.28}$ | $80.82_{\pm2.13}$ | $90.98_{\pm1.64}$ | $91.45_{\pm1.14}$ | $86.22_{\pm2.45}$ | $92.95_{\pm1.31}$ | $\mathbf{93.12_{\pm0.65}}$ |
| Cornell | $65.90_{\pm4.43}$ | $78.21_{\pm2.95}$ | $91.81_{\pm1.96}$ | $90.82_{\pm1.63}$ | $83.93_{\pm2.13}$ | $91.37_{\pm1.81}$ | $\mathbf{92.13_{\pm1.64}}$ |

use the following propagation process:

$$\mathbf{Z} = \sum_{k=0}^{K} \theta_k \frac{1}{2^K} \binom{K}{k} (2\mathbf{I} - \mathbf{L})^{K-k} \mathbf{L}^k f(\mathbf{X}), \tag{7}$$

where $f(\mathbf{X})$ is a 2-layer MLP with 64 hidden units on the feature matrix $\mathbf{X}$. Note that this propagation process is almost identical to that of APPNP or GPR-GNN. The only difference is that we substitute the Generalized PageRank polynomial with Bernstein polynomial. We set the $K = 10$ and use different learning rate and dropout for the linear layer and the propagation layer. For all models, we optimal leaning rate over $\{0.001, 0.002, 0.01, 0.05\}$ and weight decay $\{0.0, 0.0005\}$. More detailed experimental settings are discussed in Appendix.

We use the micro-F1 score with a 95% confidence interval as the evaluation metric. The relevant results are summarized in Table 4. Boldface letters indicate the best result for the given confidence interval. We observe that BernNet provides the best results on seven out of the ten datasets. On the other three datasets, BernNet also achieves competitive results against SOTA methods.

More interestingly, this experiment also shows BernNet can learn complex filters from real-world datasets with only the supervision of node labels. Figure 4 plots some of the filters BernNet learnt in the training process. On Actor, BernNet learns an all-pass-alike filter, which concurs with the fact that MLP outperforms all other baselines on this dataset. On Chameleon and Squirrel, BernNet learns two comb-alike filters. Given that BernNet outperforms all competitors by at least 1% on these two datasets, it may suggest that comb-alike filters are necessary for Chameleon and Squirrel. Figure 5 shows the Coefficients $\theta_k$ learnt from real-world datasets by BernNet. When comparing Figures 4 and 5, we observe that the curves of filters and curves of coefficients are almost the same. This is because BernNet's coefficients are highly correlated with the spectral property of the target filter, which indicates BernNet Bernnet has strong interpretability.

Finally, we present the train time for each method in Table 5. BernNet is slower than other methods due to its quadratic dependence on the degree $K$. However, compared to the SOTA method GPR-GNN, the margin is generally less than 2, which is often acceptable in practice. In theory, both ChebNet [7] and GPR-GNN [5] are linear time complexity related to propagation step $K$, but BernNet is quadratic time complexity related to $K$. Delgado et al. [8] show that Bernstein approximation can be evaluated in linear time related to $K$ using the corner cutting algorithm. However, BernNet can not use this algorithm directly, because we need to multiply signal $\mathbf{x}$. How to convert BernNet to linear complexity will be a problem worth studying in the future.

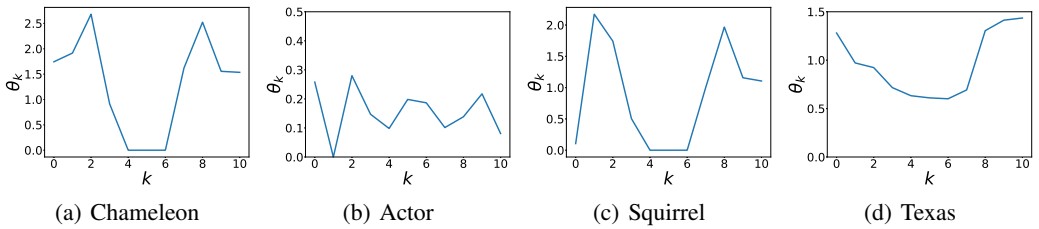

(a) Chameleon   (b) Actor   (c) Squirrel   (d) Texas

Figure 5: Coefficients $\theta_k$ learnt from real-world datasets by BernNet.

Table 5: Average running time per epoch (ms)/average total running time (s).

|  | GCN | GAT | APPNP | MLP | ChebNet | GPR-GNN | BernNet |
|---|---|---|---|---|---|---|---|
| Cora | 4.59/1.62 | 9.56/2.03 | 7.16/2.32 | 3.06/0.93 | 6.25/1.76 | 9.94/2.21 | 19.71/5.47 |
| CiteSeer | 4.63/1.95 | 9.93/2.21 | 7.79/2.77 | 2.95/1.09 | 8.28/2.56 | 11.16/2.37 | 22.36/6.32 |
| PubMed | 5.12/1.87 | 16.16/3.41 | 8.21/2.63 | 2.91/1.61 | 18.04/3.03 | 10.45/2.81 | 22.02/8.19 |
| Computers | 5.72/2.52 | 30.91/7.85 | 9.19/3.48 | 3.47/1.31 | 20.64/9.64 | 16.05/4.38 | 28.83/8.69 |
| Photo | 5.08/2.63 | 19.97/5.41 | 8.69/4.18 | 3.67/1.66 | 13.25/7.02 | 13.96/3.94 | 24.69/7.37 |
| Chameleon | 4.93/0.99 | 13.11/2.66 | 7.93/1.62 | 3.14/0.63 | 10.92/2.25 | 10.93/2.41 | 22.54/4.75 |
| Actor | 5.43/1.09 | 11.94/2.45 | 8.46/1.71 | 3.82/0.77 | 7.99/1.62 | 11.57/2.35 | 23.34/5.81 |
| Squirrel | 5.61/1.13 | 22.76/4.91 | 8.01/1.61 | 3.41/0.69 | 38.12/7.78 | 9.87/5.56 | 25.58/9.23 |
| Texas | 4.58/0.92 | 9.65/1.96 | 7.83/1.63 | 3.19/0.65 | 6.51/1.34 | 10.45/2.16 | 23.35/4.81 |
| Cornell | 4.83/0.97 | 9.79/1.99 | 8.23/1.68 | 3.25/0.66 | 5.85/1.22 | 9.86/2.05 | 22.23/5.26 |

# 6   Conclusion

This paper proposes *BernNet*, a graph neural network that provides a simple and intuitive mechanism for designing and learning an arbitrary spectral filter via Bernstein polynomial approximation. Compared to previous methods, BernNet can approximate complex filters such as band-rejection and comb filters, and can provide better interpretability. Furthermore, the polynomial filters designed and learned by BernNet are always valid. Experiments show that BernNet outperforms SOTA methods in terms of effectiveness on both synthetic and real-world datasets. For future work, an interesting direction is to improve the efficiency of BernNet.

## Broader Impact

The proposed BernNet algorithm addresses the challenge of designing and learning arbitrary spectral filters on graphs. We consider this algorithm a general technical and theoretical contribution, without any foreseeable specific impacts. For applications in bioinformatics, computer vision, and natural language processing, applying the BernNet algorithm may improve the performance of existing GNN models. We leave the exploration of other potential impacts to future work.

## Acknowledgments and Disclosure of Funding

Zhewei Wei was supported in part by National Natural Science Foundation of China (No. 61972401, No. 61932001 and No. 61832017), by Beijing Outstanding Young Scientist Program NO. BJJWZYJH012019100020098, by Alibaba Group through Alibaba Innovative Research Program, and by CCF-Baidu Open the Fund (NO.2021PP15002000). Zengfeng Huang was supported by National Natural Science Foundation of China Grant No. 61802069, and by Shanghai Science and Technology Commission Grant No. 17JC1420200. Hongteng Xu was supported by Tencent AI Lab Rhino-Bird Joint Research Program. This work is supported by China Unicom Innovation Ecological Cooperation Plan and by Intelligent Social Governance Platform, Major Innovation & Planning Interdisciplinary Platform for the "Double-First Class" Initiative, Renmin University of China. We also wish to acknowledge the support provided and contribution made by Public Policy and Decision-making Research Lab of Renmin University of China.

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
