# OpenReview forum: "BernNet: Learning Arbitrary Graph Spectral Filters via Bernstein Approximation"
_NeurIPS.cc/2021/Conference — NeurIPS 2021 Poster_

### Official Review · Reviewer_VEJH · 2021-07-11

**Rating:** 7
**Confidence:** 5

**Summary:**

The authors propose BernNet, which uses Bernstein polynomial basis to learn graph filters. From extensive experiments and theoretical reasoning, the authors demonstrate that BernNet is favored over baselines with a similar idea such as GPR-GNN and ChebyNet.

**Limitations And Societal Impact:**

Limitations on computational complexity are addressed.

**Main Review:**

Strengths:

(+) The theoretical motivation of using the Bernstein basis is neat and inspiring.

(+) The synthetic experiments on the ability of BernNet to learn arbitrary graph filters well support its validity.

(+) The paper is well written.

(+) The higher computational complexity of BernNet over simpler models such as GPR-GNN is clearly stated. So, the performance-complexity trade-off is clear to the readers.

Minor Weakness:

(-) Do not compare with the other popular polynomial basis such as Chebyshev (different from ChebyNet), Legendre. I understand that they might not have the same good properties as Bernstein, but some synthetic experiments would be appreciated.

Detail comments:

Overall, I like the main idea of this paper. The motivation of why we should use Bernstein basis is clear and neat. The synthetic experiments on learning filters for images are interesting and convincing. Although the model itself just changes monomial basis in GPR-GNN to Bernstein basis, the idea and theoretical reasoning behind it is profound.

Still, I think some minor things can further improve the current manuscript. The authors clearly show that it is reasonable to consider the non-negative constraint and thus Bernstein basis becomes a great choice. However, when it comes to polynomial approximation the other popular polynomial basis should also be considered, such as Chebyshev and Legendre. Note that using Chebyshev basis in GPR-GNN is not the same as ChebyNet. I think the current manuscript can be even more solid if such an experiment can be done.

Originality: Great. Although the model itself does not differ a lot, the idea and motivation behind it are inspiring.

Quality: Both theoretical reasoning and experiments are convincing.

Clarity: The paper is well written. Maybe the importance of the non-negative graph filters section can be further emphasized.

Significance: This work has moderate significance.


**Time Spent Reviewing:**

5

---

> ### Author Response · Authors · 2021-08-10
> **Our Response**
>
> Thank you for your insightful feedback.
>
> **Q1:** Compare with the other popular polynomial basis such as Chebyshev (different from ChebyNet), Legendre.
>
> **A1:** First of all, Bernstein approximation has a nice non-negative property that Chebyshev approximation or any other polynomial basis does not: By setting $\theta_k \ge 0$ for $k=0,\ldots, K$, Bernstein approximation produces only non-negative polynomials on $[0,2]$ (see Lemma 2.1 in the paper). On the other hand, any degree $K$ polynomial that is not negative on $[0,2]$ can be expressed as a Bernstein approximation with a non-negative $\theta_k$ (see Lemma 3.1 in the paper). This means that we can force Bernstein polynomials to approximate non-negative graph filters and non-negative graph filters only. On the other hand, Chebyshev polynomial approximation and Legendre polynomial approximation may generate filters with a negative response.
>
>
> Non-negativity, we believe, is critical for designing valid graph spectral filters. 1) From the system stability or robustness standpoint, the negative response causes the output result to oscillate as the number of layers is changed between odd and even. It is unsuitable for deep graph neural network design and implementation. 2) As demonstrated in Section 3, the negative response degrades the semi-positive definite nature of $\gamma(\mathbf{L})$, causing the optimization function to converge to a saddle point or even non-convergence. 3) The frequency response of the filter between $[0,1]$ is the most common condition for a variety of filters used in signal processing, such as Butterworth and Chebyshev. When the frequency response is less than zero, the signal produces an oscillatory output, while a frequency response greater than one results in signal amplification, both of which can result in system instability when cascading. As a result, our goal is to design and learn non-negative polynomial filters, which BernNet accomplishes perfectly.
>
> Additionally, as discussed in Section 2.3, Bernstein polynomial approximation provides superior interpretability and controllability when designing and learning graph filters. The non-negative coefficients $\theta_k, k=0,\ldots, K$ can be interpreted as $h(2k/K)$, $k=0, \ldots, K$, which correspond directly to the filter values sampled uniformly in $[0,2]$. By comparing Figures 5 and 6 in the supplemental material, we can see that the shape of $\theta_k$'s is nearly identical to that of the filter $h$, demonstrating BernNet's interpretability.
>
> Finally, as suggested, we present in Table 1 the comparative studies of BernNet with Chebyshev and Legendre polynomial approximation on five datasets. Chebyshev and Legendre are identical to BernNet except that the Bernstein polynomial approximation is replaced by Chebyshev and Legendre polynomial approximations, respectively. BernNet continues to outperform Chebyshev and Legendre, which we believe is because 1) BernNet always guarantees the non-negativity of the learned filter, and 2)  BernNet provides superior interpretability and controllability in terms of the learned coefficients $\theta_k$'s.
>
>
> **Table 1: Results on real-world benchmark datasets: Mean accuracy (\%) ± 95\% confidence interval.**
>
> |           |  Computers |    Photo   |   Chamelon  |    Actor   |    Texas   |
> |-----------|:----------:|:----------:|:-----------:|:----------:|:----------:|
> | Chebyshev | 86.73±2.11 | 93.15±0.59 | 66.85±2.27  | 39.79±0.94 | 83.61±3.41 |
> | Legendre  | 87.18±0.58 | 93.41±0.46 |  67.41±2.62 | 39.47±0.67 | 88.52±4.56 |
> | BernNet   | **87.64±0.44** | **93.63±0.35** |  **68.29±1.58** | **41.79±1.01** | **93.12±0.65** |
>
>
> We will gladly answer any additional questions you may have.

---

> > ### Comment · Reviewer_VEJH · 2021-08-25
> > **Re**
> >
> > Thanks for the response and additional experiments! I have no more problem with this paper. Wish the authors the best of luck.

---

### Official Review · Reviewer_ZSJK · 2021-07-16

**Rating:** 6
**Confidence:** 4

**Summary:**

This paper proposes a new graph neural network (GNN) architecture whose convolutional layer is capable of learning spectral graph filters of arbitrary characteristics via Bernstein approximation. Under the constraint of positive filter response, the authors show that the polynomial which approximates the solution to a generalised graph-based optimisation necessarily has the form of a non-negative linear combination of Bernstein basis, which justifies the proposed GNN convolutional layer. Experimental results demonstrate that the proposed scheme can learn complex filters and achieve competitive results on a number of benchmark datasets.

**Limitations And Societal Impact:**

Limitations are mainly discussed from the perspective of computational complexity for which running time results have been reported and discussed.

**Main Review:**

This paper proposes a new GNN architecture following the so-called spectral designs. Compared to spatial designs (e.g. message-pass neural networks), spectral designs have been arguably less extensively studied, and in this sense the paper adds value to the GNN literature. The proposed graph convolutional layer involves a filter based on Bernstein polynomial approximation and has several advantages: 1) it guarantees a non-negative filter response; 2) it can adapt to complex filter characteristics; 3) it is theoretically justified via a graph-based optimisation problem. Despite the rich literature, these are interesting aspects that enrich the design of GNN architectures.

The paper is generally technically sound and ideas are presented in a clear and coherent manner. The illustrations of the filter response of various types are helpful for understanding the advantage of the proposed approach.

My main concern is that the advantages of the proposed approach over existing methods that can learn flexible spectral filters (e.g. ChebNet) should be better justified. Theoretically, the authors should explain in more depth why the Bernstein polynomial approximation is potentially better than the one by Chebyshev polynomials. Empirically, in addition to performance metrics, it would be good to compare the learned filters from BertNet and ChebNet (among others) so that their differences can be made more clear.

Another major issue is that, to increase the expressivity of the proposed filter we need to increase the order of the polynomial K, which increases the computational complexity quadratically. This can in certain scenarios be a significant shortcoming compared to approaches such as ChebNet whose complexity is linear with respect to K.

Overall, I am slightly inclined to recommend acceptance due to the positive aspects of the paper mentioned above, but I would recommend the authors address some of the issues raised in the review during rebuttal.

Below are additional comments that may help further improve the quality of the paper:

The statement “none of the existing work supports designing arbitrary spectral filters” is not precise as architectures such as ChebNet can do that in theory.

I am not sure it is the negative spectral response that leads to the over-smoothing of GNNs as claimed by the authors. To me these seem to be separate issues (unless a definitive link can be made which would be welcome).

The claim that some existing GNNs may not satisfy Constraint 3.1 is in my view not precise as it depends on the range of spectrum. For example, the GCN filter would guarantee non-negative filter response if the eigenvalues are upper bounded by 1, which is the case for scaled or shifted normalised graph Laplacian (which are commonly used in the literature, e.g. in ChebNet). This corresponds to one of the main novelties of the paper and needs to be explained in more depth. Related to a comment above, it would be good to compare the learned filters using BernNet and baselines to demonstrate this point.

The statement “these methods mainly focus on designing low- or high-pass filters…” is not entirely precise. In theory, as long as the architecture can adapt to data characteristics (e.g. non-smoothness) they can learn flexible filters beyond low- or high-pass ones.

In 5.1, the formulation of the problem as a regression problem should be explained more clearly (e.g. what do the authors means by using “a mask to cover the edge nodes”?).

Some recent work on learning filters of various spectral characteristics should be discussed in Related Work, for example:
- https://arxiv.org/abs/2011.10988
- https://arxiv.org/abs/2104.12840

------post-rebuttal------

I thank the authors for their response and clarifications. I would suggest that the authors further improve their paper based on these discussions. One thing I am not sure about is the authors' claim that it would not be easy to check the filter response from ChebNet. In my view this should be possible to do (see for example a recent work https://arxiv.org/abs/2003.11702) and can further strengthen the paper.


**Time Spent Reviewing:**

3

---

> ### Author Response · Authors · 2021-08-10
> **Our Response**
>
> Thank you for your insightful feedback. We answer your questions below.
>
> **Q1:** The advantages of the proposed approach over existing methods that can learn flexible spectral filters (e.g. ChebNet) should be better justified.
>
> **A1:** First of all, Bernstein approximation has a nice non-negative property that Chebyshev approximation or any other polynomial basis does not: By setting $\theta_k \ge 0$ for $k=0,\ldots, K$, Bernstein approximation produces only non-negative polynomials on $[0,2]$ (see Lemma 2.1 in the paper). On the other hand, any degree $K$ polynomial that is not negative on $[0,2]$ can be expressed as a Bernstein approximation with a non-negative $\theta_k$ (see Lemma 3.1 in the paper). This means that we can force Bernstein polynomials to approximate non-negative graph filters and non-negative graph filters only. ChebNet and GPR-GNN, on the other hand, may generate filters with a negative response.
>
> Non-negativity, we believe, is critical for designing valid graph spectral filters. 1) From the system stability or robustness standpoint, the negative response causes the output result to oscillate as the number of layers is changed between odd and even. It is unsuitable for deep graph neural network design and implementation. 2) As demonstrated in Section 3, the negative response violates the semi-positive definite nature of $\gamma(\mathbf{L})$, causing the optimization function to converge to a saddle point or even non-convergence. 3) The frequency response of the filter between $[0,1]$ is the most common condition for a variety of filters used in signal processing, such as Butterworth and Chebyshev. When the frequency response is less than zero, the signal produces an oscillatory output, while a frequency response greater than one results in signal amplification, both of which can result in system instability when cascading. As a result, our goal is to design and learn non-negative polynomial filters, which BernNet accomplishes perfectly.
>
> Additionally, as discussed in Section 2.3, Bernstein polynomial approximation provides superior interpretability and controllability when designing and learning graph filters. The non-negative coefficients $\theta_k, k=0,\ldots, K$ can be interpreted as $h(2k/K)$, $k=0, \ldots, K$, which correspond directly to the filter values sampled uniformly in $[0,2]$. By comparing Figures 5 and 6 in the supplemental material, we can see that the shape of $\theta_k$'s is nearly identical to that of the filter $h$, demonstrating BernNet's interpretability.
>
> Finally, we present Tables 1 and 2, which compare the filter responses learned by BernNet and GPR-GNN on the Actor and Squirrel datasets, respectively. Note that we cannot plot the filters learned by ChebNet because their polynomial coefficients are implicitly encoded in the weight matrices $\mathbf{W}_k$ (ChebNet's expression is $\mathbf{Z}=\sum\nolimits _{k=0}^KT_k(\tilde{\mathbf{L}})\mathbf{X}\mathbf{W}_k$  [12]). On Squirrel, GPR-GNN learns a high-pass filter that contains negative responses, while BernNet learns a comb-alike filter. On Actor, GPR-GNN learns a band-rejection filter while BernNet learns an all-pass-alike filter.  Given that MLP outperforms GPR-GNN on this dataset (see Table 4 in the paper), we believe BernNet learns a more effective filter than GPR-GNN does.
>
> **Table 1: The filter responses learned by BernNet and GPR-GNN on Actor.**
>
> | $\lambda$ |  0.0 |  0.2 |  0.4 |  0.6 |  0.8 |  1.0 |  1.2 |  1.4 |  1.6 |  1.8 |  2.0 |
> |-----------|:----:|:----:|:----:|:----:|:----:|:----:|:----:|:----:|:----:|:----:|:----:|
> | GPR-GNN   | 0.89 | 0.71 | 0.69 | 0.67 | 0.65 | 0.64 | 0.66 | 0.71 | 0.77 | 0.89 | 1.22 |
> | BernNet   | 0.25 | 0.15 | 0.16 | 0.16 | 0.16 | 0.16 | 0.16 | 0.15 | 0.15 | 0.15 | 0.08 |
>
>
>
> **Table 2: The filter responses learned by BernNet and GPR-GNN on Squirrel.**
>
> | $\lambda$ |  0.0 |  0.2 |  0.4 |  0.6 |  0.8 |  1.0 |  1.2 |  1.4 |  1.6 |  1.8 |  2.0 |
> |-----------|:----:|:----:|:----:|:----:|:----:|:----:|:----:|:----:|:----:|:----:|:----:|
> | GPR-GNN   | 0.36 | -0.18 | -0.11 | 0.12 | 0.27 | 0.25 | 0.07 | -0.12 | 0.01 | 1.35 | 6.33 |
> | BernNet   | 0.11 | 1.25 | 1.22 | 0.82 | 0.47 | 0.37 | 0.55 | 0.91 | 1.22 | 1.27 | 1.11 |
>
>
> **Q2:** The computational complexity quadratically of BernNet can, in certain scenarios, be a significant shortcoming.
>
> **A2:** You are correct. BernNet has an $O(K^2)$ time complexity, whereas ChebNet and GPR-GNN are only linearly dependent on $K$. However, as shown in Table 5 in the paper, when $K=10$, there is no significant difference in the running time of BernNet, ChebNet, and GPR-GNN. As discussed in Section 5.2, we believe that developing a linear algorithm for BernNet is an intriguing future direction. Delgado et al.[8] demonstrates that the Bernstein polynomial can be evaluated in linear time in terms of $K$ by employing the corner-cutting algorithm. BernNet, on the other hand, cannot use this algorithm directly because it requires multiplying signal $\mathbf{x}$.
>
> **Additional comments:** We appreciate the advice! We will make the following changes to the final version of the paper: 1) We will change the corresponding statement to "no existing work supports the design of arbitrary non-negative spectral filters." 2) We will remove the statement that associates over-smoothing with negative filters; 3) We will compare the learned filters from BernNet to those from GPR-GNN and other methods; 4) We will modify the statement to read "these methods are primarily concerned with designing or learning filters without regard for the non-negativity constraint, which may result in misspecified or even ill-posed filters." 5) We will present a detailed discussion of these experiments. In essence, the setting is identical to that of [1]; 6) The suggested works will be incorporated into the final version of the paper.
>
> We will gladly answer any additional questions you may have.

---

### Official Review · Reviewer_UYuG · 2021-07-17

**Rating:** 8
**Confidence:** 5

**Summary:**

This paper proposes BernNet, a novel graph neural network with theoretical support that provides a simple but effective scheme for designing and learning arbitrary graph spectral filters. The key property of BernNet is that 1) any valid polynomial filers that maps [0,2] to [0,1] can always be expressed by a BernNet, and 2) the filters learned by our BernNet are always non-negative. The experiments demonstrate that BernNet can learn complex spectral filters, such as band-rejection and comb filters, and it achieves superior performance in real-world graph modeling tasks. Overall, the motivation is clear and the idea is interesting. Technical contributions are clearly shown with both theoretical and empirical evidence.

**Limitations And Societal Impact:**

Yes, the authors addressed all these aspects.

Describe the limitations of your work? [Yes] BernNet is quadratic dependent on K.

Any potential negative societal impacts of this work? [Yes] see Section 7 [Broader Impact]

**Main Review:**

This work is mostly well-written with informative content and new ideas that are worth publishing.

Strengths:

1. The use of the Bernstein basis to approximate an arbitrary graph filter is an exciting idea.

2. The Bernstein basis has higher interpretability and controllability than the Chebyshev basis since its coefficients directly match the filter values.

3. The authors demonstrate that any non-negative polynomial filter can be expressed using a non-negative linear combination of the Bernstein basis, distinguishing Bernstein from other learnable GNNs (e.g. ChebNet, GPR).

4. The experiments reveal some intriguing findings. BernNet, for example, learns an MLP from the Actor dataset, which seems to outperform other advanced GNN models.

Weakness:

1. The technical section of the paper is relatively dense, which may be difficult for non-experts in GNN to understand.

2. From the standpoint of graph optimization, the authors justify the non-negative feature of polynomial filters. While this motivation makes perfect sense, it appears that a valid filter has to be non-negative by definition.


**Time Spent Reviewing:**

48

---

> ### Author Response · Authors · 2021-08-10
> **Our Response**
>
> We appreciate your insightful feedback.
>
> **Q1:** The technical section of the paper is relatively dense, which may be difficult for non-experts in GNN to understand.
>
> **A1:** We will revise the text and include some context information in the final version of the paper to assist readers in comprehending the technical section.
>
> **Q2:** While the motivation of the non-negative filter justified  from the graph optimization makes sense, it appears that a valid filter has to be non-negative by definition.
>
> **A2:** We fully agree that a valid filter must be a non-negative function. Indeed, BernNet's primary motivation has been to construct arbitrary non-negative graph filters. The graph optimization section provides additional evidence for why the graph filters in graph neural networks must be non-negative. We considered three aspects when determining whether the filter must produce a non-negative response. To begin, the negative response will cause the output result to oscillate as the GNN depth changes. It is unsuitable for deep graph neural network design and implementation. Second, the negative response destroys the semi-positive definite quality of $\gamma(\mathbf{L})$, causing the optimization function to converge to a saddle point or even to non-convergence. Finally, the frequency response of the filter between $[0,1]$ is the standard condition for a variety of filters used in signal processing, such as Butterworth and Chebyshev. When the frequency response is less than zero, the signal produces an oscillatory output, while a frequency response greater than one results in signal amplification, both of which can result in system instability when cascading. We may be able to provide clearer and easier-to-understand theoretical guarantees from the perspective of graph optimization functions. It is self-evident that using the graph optimization function, one can verify that the filter has a non-negative response (Equation 4 in Section 3).
>
> We will gladly answer any additional questions you may have.

---

> > ### Comment · Reviewer_UYuG · 2021-08-26
> > **Response to Authors**
> >
> > Thank you for your response. All my concerns have been clarified. I have no further comments on this paper.

---

### Decision · Program_Chairs · 2021-09-27

**Decision:**

Accept (Poster)

**Comment:**

The paper proposes a new algorithmic approach for GNNs, and is backed up with both theory and empirical validation. The reviewers are all generally positive and in agreement of its important contributions.